# Chemistry of Substituted Thiazinanes and Their Derivatives

**DOI:** 10.3390/molecules25235610

**Published:** 2020-11-28

**Authors:** Alaa A. Hassan, Stefan Bräse, Ashraf A. Aly, Hendawy N. Tawfeek

**Affiliations:** 1Chemistry Department, Faculty of Science, Minia University, El-Minia 61519, Egypt; ashrafaly63@yahoo.com (A.A.A.); Hendawy.Nagaty_pg@sci.s-mu.edu.eg (H.N.T.); 2Institute of Organic Chemistry, Karlsruhe Institute of Technology, 76131 Karlsruhe, Germany; 3Institute of Biological and Chemical Systems (IBCS-FMS), Karlsruhe Institute of Technology, 76344 Eggenstein-Leopoldshafen, Germany

**Keywords:** biologic activity, fused-heterocycles, spiro compounds, structures, thiazinanes

## Abstract

Thiazinanes and its isomeric forms represent one of the most important heterocyclic compounds, and their derivatives represented a highly potent drug in disease treatment such as, 1,1-dioxido-1,2-thiazinan-1,6-naphthyridine, which has been shown to have anti-HIV activity by a mechanism that should work as anti-AIDS treatment, while (*Z*)-methyl 3-(naphthalen-1-ylimino)- 2-thia-4-azaspiro[5 5]undecane-4-carbodithioate showed analgesic activity, cephradine was used as antibiotic and chlormezanone was utilized as anticoagulants. All publications were interested in the chemistry of thiazine (partially or fully unsaturated heterocyclic six-membered ring containing nitrogen and sulfur), but no one was dealing with thiazinane itself which encouraged us to shed new light on these interesting heterocycles. This review was focused on the synthetic approaches of thiazinane derivatives and their chemical reactivity.

## 1. Introduction

Nitrogen–sulfur containing heterocycles represent a widespread group of heterocyclic compounds. These types of heterocycles constructed a large number of drugs used in the treatment of a variety of diseases. Thiazinane resembles a compound containing nitrogen and sulfur on its structure. It is a fully saturated thiazine six-membered ring containing two hetero-atoms nitrogen and sulfur in a three isomeric structures [1,2]thiazinane, [1,3]thiazinane and [1,4]thiazinane as mentioned below (Figure 1).

1,3-Thiazine framework represented an important structural motif presented in natural products (bretschneiderazines A & B) [1] (Figure 2) and bioactive compounds [2,3,4]. The well-known antibiotics, cephamycin and cephradine (cephalosporin class of β-lactam antibiotics) containing a 1,3-thiazine skeleton [2] (Figure 2).

In addition, several synthetic 1,3-thiazine derivatives shown various biologic activities such as analgesic [3], antihypotensive [4] and NOS (nitric oxide synthases) inhibiting activities (Figure 2) [4].

1,1-Dioxido-1,2-thiazinan-1,6-naphthyridine is an HIV integrase inhibitor currently undergoing evaluation for the treatment of AIDS (acquired immune deficiency syndrome) (Figure 3) [5].

Chlormezanone (Figure 3) is a centrally acting muscle relaxant [6]. It was introduced into human therapy as a racemic monosubstance, later also in combination with codeine phosphate and paracetamol. Chloromezanone (Figure 3) was widely used as anticoagulant [7]. Other derivatives have shown wide range activities as antimicrobial [8,9] and peptic ulcer treatment [10] and anti-inflammatory [11].

Eflornithine (α-difluoromethylornithine), an ornithine decarboxylase inhibitor, is active against second-stage Trypanosoma brucei gambiensis [12] and has been used in conjunction with nifurtimox against Trypanosoma brucei [13,14] (Figure 4). In addition, 2-nitromethylene-1,3-thiazinan-3-yl- carbam-aldehyde was used as an insecticide [15] (Figure 4).

On the other hand, thiazinanones, are very interesting compounds due to their important role in medicinal chemistry [16,17,18]. It has been reported that, substituted thiazinanones exhibited antitumor [19], antifungal activity [20] and antimalarial activity [21], as well as antioxidant activity [22]. Reactions of amine, carbonyl compounds and a mercapto acid in one-pot three-component condensation or a two-step process afforded thiazinanone derivatives [20].

## 2. Chemistry of Thiazinanes

### 2.1. Synthesis of Thiazinanes

#### 2.1.1. Synthesis of 1,2-thiazinanes

1,2-Thiazinane-1,1-dioxide derivatives **5a**–**d** (yields 10–50%) and **6a**–**d** (yields 22–28%) as diastereoisomers were synthesized from the corresponding amino-halides **1** or amino-alcohols **2**. The sultam rings were constructed according to the method of Lee et al. [23] Compounds **1** were reacted with phenylmethanesulfonyl chloride in presence of triethylamine (Et_3_N) gave the secondary sulfonamides, treatment with base facilitate cyclization to the sultam ring intermediates **3**. Similar to **1,** derivatives of compound **2** were reacted with phenylmethanesulfonyl chloride and triethylamine, followed by treatment with NaCl yielded the alkyl bromide intermediates. The latter were treated with a base gave the sultam ring intermediates **3**. Treatment of **3** with sodium hydride and 4-bromo-1-(bromomethyl)-2-fluorobenzene gave *N*-benzyl sultam intermediates **4**. Intermediates **4** were subjected to Buchwald–Hartwig amination by reacting 2-dicyclohexyl phosphino-2′,6′-diisopropoxybiphenyl (RuPhos) (as a reagent in palladium-catalyzed cross-coupling) [24] with *N*-acetylpiperazine to give the sultam products as mixtures of enantiomers and diastereomers **5** and **6**, which has been separated using chiral supercritical fluid chromatography (SFC) (Scheme 1) [11].

Homo-allylic sulfamate ester **7** and sulfonamide **9** were useful substrates for the *T*ethered *A*minohydroxylation (TA) reaction. The sulfamate ester **7** was underwent the TA reaction giving 1,2,3-oxathiazinane product **8** (yields 53–68%). In contrast, the sulfonamide (pent-4-ene-1- sulfonamide) **9** gave 1,2-thiazinane product (1,1-dioxo-[1,2]thiazinan-3-yl) methanol) **10** (yields 35–59%) under the same conditions (Scheme 2) [25].

But-3-ene-1-sulfonamide **11** underwent intramolecular aziridination to give the bicyclic aziridines **12**. Reaction of 5-hexenyl-substituted sulfonamide **14** only furnished the product derived from allylic insertion 3-vinyl [1,2]thiazinane-1,1-dioxide **15** (yield 70%). Treatment of azabicyclic sulfonamide **12** (2-thia-1-azabicyclo-[3,1,0]hexane-2,2-dioxide) with *p*-toluenesulfonic acid (*p*-TsOH) resulted in ring-opening of the aziridine **12** at the more substituted position affording the six-membered ring product 4-methoxy-1,2-thiazinane-1,1-dioxide (**13**) (yield 60%). The aziridination ring-opening was facilitated in the presence of Lewis acid (Scheme 3) [26].

Unsaturated sulfonamide (hex-5-ene-1-sulfonamide) (**14**) underwent intramolecular aziridination catalyzed by Rh_2_(OAc)_4_ with PhI(OAc)_2_ and Al_2_O_3_ to give the corresponding 3-vinyl-1,2-thiazinane-1,1-dioxide (**15**) (yield 90%) (Scheme 4) [27].

Reactions of ethyl 2-(chlorosulfonyl) acetate (**16**) with amines furnished sulfonamides **17a**–**c**. Upon treatment of **17a**–**c** with 1-bromo-3-chloropropane in DMF and in presence of K_2_CO_3_ gave the six-membered cyclic sulfamoyl acetamide esters (ethyl 2-aryl-1,2-thiazinane-6-carboxylate-1,1-di-oxide) **18a**–**c**. Hydrolysis of **18a**–**c** using methanolic KOH gave **19a**–**c**. Coupling of **19a**–**c** with 4-(4-amino-2-fluorophenoxy)-3-chloropicolinamide (**20**), under 1-ethyl-3-(3-dimethylaminopropyl) carbodiimide (EDC)/HCl and *N*,*N*-dimethylpyridin-4-amine (DMAP) conditions in THF yielding intermediates 2-substituted-1,2-thiazinane-6-carboxamide-1,1-dioxide **21a**–**c** (yields 43–55%). Compound **21a**–**c** underwent Hoffman rearrangement using iodobenzenediacetate furnished 2-amino-3-chloropyridin 2-substituted-1,2-thiazinane-6-carboxamide-1,1-dioxides **22a**–**c** in yields 58–68% (Scheme 5) [28].

In addition, the isomeric six-membered sulfamoyl acetamides **27a**–**c** were obtained from coupling between chloroacetyl chloride and substituted anilines to give compounds **23a**–**c**, which were converted to sulfamoyl chlorides **24a**–**c** in the presence of sodium sulfite followed by phosphorous pentachloride (PCl_5_). Coupling of substituted anilines (4-(4-amino-2-fluorophenoxy) -3-chloropicolinamide) **20a**–**c** with sulfamoyl chlorides **24a**–**c** gave sulfamoyl acetamides **25a**–**c** in presence of *N*,*N*-diisopropylethylamine (DIPEA) in dry THF. Treatment of **25a**–**c** with 1,3-bromochloropropane in the presence of potassium carbonate gave cyclic sulfamoyl acetamides **26a**–**c** (yields 52–59%). Hoffman rearrangement in compounds **26a**–**c** using PhI(OAc)_2_ as a mediator yielded sulfamoyl acetamides **27a**–**c** in moderate yields 58–68% (Scheme 6) [28].

*ω*-Alkene-1-sulfonamides **14a**–**d** was prepared by aminolysis of *ω*-alkene-1-sulfonyl chlorides **28a**–**d**. Allylsulfonamide (**29a**) did not lead to the highly strained bicyclic [2.1.0] structure **30**. In contrast, the higher homologues **14b,c** gave bicyclic aziridines **31** and **32**, respectively. However, sulfonamide **28d** under the same conditions gave rise to the allylic insertion product (3-vinyl-1,2-thiazinane-1,1-dioxide) **15**. Using different types of nucleophiles (alcohol, thiophenol, allyl magnesium bromide, benzylamine) afforded aziridine ring-opened products in good yields with C–O, C–S, C–C or C–N bond formation. Ring-opening of the aziridine at the more substituted site take place in case of compounds **31** and **32**, leading to six- and seven-membered ring products 4-methoxy-1,2-thiazinane-1,1-dioxide (**33a**) (Yield 65%), 4-(phenylthio)-1,2-thiazinane-1,1-dioxide (**33b**) (Yield 62%) and 4-methoxy-1,2-thiazepan-1,1-dioxide (**34**) (yield 92%), respectively using copper (I) or (II) trifluoromethanesulfonate (Cu (I or II) OTf) and sodium hydride as reagents (Scheme 7) [29].

4-(4-Bromo-3,5-dimethylphenoxy)-1,2-thiazinane-1,1-dioxide (**35**) was prepared in 66% yield, from the reaction between 2-thia-1-aza-bicyclo[3.1.0]hexane-2,2-dioxide (**31**) and 4-bromo-3,5-dimethylphenol in *N*,*N*-dimethylacetamide (DMAc) via ring opening–ring closure interaction. The thiazinane **35** when treated with NaH in *N*,*N*-dimethylacetamide and iodomethane gave 4-(4-bromo-3,5-dimethylphenoxy)-2-methyl[1,2]thiazinane-1,1-dioxide (**36**) (yield 32%) (Scheme 8) [30].

The sulfonamide (*N*,*N*-bis(4-methoxybenzyl)methanesulfonamide) (**37**) was treated with lithium hexamethyldisilazide (LiHMDS), followed by addition of diethyl chlorophosphate and quenched with 5-bromo-2-methoxybenzaldehyde to form alkenyl sulfonamide **38** in 80% yield. Compound **38** was subjected to Michael-addition using dimethyl malonate to form the diester. Decarboxylation and sulfonamide deprotection of **38** formed the sulfonamide **39** (yield 46%). Cyclisation of **39** using standard NaOMe furnished 5-aryl-1,2-thiazinan-3-one-1,1-dioxide **40** in good yield 74%, after Suzuki coupling with phenylboronic acid (Scheme 9) [31].

Terminal alkenes and hydroamination of inactivated alkenes have been isomerized using phosphine gold (I) complexes as a catalyst under both thermal and microwave conditions. Sulfonamides **14a**,**b** readily underwent intramolecular hydroamination to give thiazinane-1,1-dioxides **41a**,**b** (yields 95% and 88%), respectively (Scheme 10) [32].

#### 2.1.2. Syntheses of 1,3-thiazinanes

##### Syntheses of N-tosyl-1,3-thiazinanes

*N*-Tosyldiazoketamine **42** was converted to the corresponding *E* (5%)/*Z* (95%)-*α*-phenyl-*β*-enamino ester **43** via decomposition of **42** through losing of N_2_ to form carbine followed by 1,2-phenyl migration under two different catalytic conditions, Rh_2_(OAc)_4_ and *p*-TsOH. For the reaction catalyzed by Rh_2_(OAc)_4_, *E*-isomer **43b** (91%) was found to be the major product along with the formation of very small quantities of the *Z*-isomer of 1,2-phenyl migration product **43a** (5%) and 1,2-hydride migration product **44** (4%). The ratio of **43a**/**43b**/**44** was found to be 5:91:4. In contrast, the 1,2-hydride migration product **44** could not be detected in reactions catalyzed by *p*-TsOH. Moreover, in the latter case, the *Z*-*α*-phenyl-*β*-enamino ester **43a** was formed as the major product (**43a**/**43b** = 95:5). Mitsunobu adduct **46** was obtained via premixing DEAD (Diethyl azodicarboxylate) and PPh_3_, followed by addition of *Z*-α-phenyl-*β*-enamino ester **43a** and alcohol **45**. The cyclized products **47** (yield 79%) were obtained from alkenylthiols **46** in one pot using trifluoroacetic acid (TFA) in diastereoselectivities (86:14) (Scheme 11) [33].

##### Synthesis of Epipyridazinoanthracen-1,3-thiazinane Propanenitrile

Reaction between thiocarbamoyl derivative **48** and 1,3-dibromopropane in presence of Et_3_N furnished the stereoselective product cyclic ketene *S*,*N*-acetal ((*E*)-3-((9*s*,10*s*)-12,15-dioxo- 9,11,12,14,15,16-hexahydro-9,10-[4,5]epipyridazinoantracen-13(10*H*)-yl)-3-oxo-2-(3-phenyl-1,3-thiazinan-2-ylidene)propanenitrile) (**49**) in 70% yield (Scheme 12) [34].

Cornia et al. [35] utilized Berzelius reagent P_4_S_10_ (phosphorus decasulfide or phosphorus pentasulfide) for thionation. 3-Hydroxypropane amide **50** combined with hexamethyldisiloxane (HMDO) gave thioamide **51**. Cyclization of intermediate **51** to 1,3-thiazine **52** (59%), which acylated using 2,2-dichloropropanoyl chloride to give (*Z*)-2,2-dichloro-1-(2-propylidene-1,3-thiazinan-3-yl) butan-1-one **54a** and (*Z*)-1-(2-benzylidene-1,3-thiazinan-3-yl)-2,2-dichloropropan-1-one **54b** in 56% and 90% yield, respectively. In addition, 2-ethyl-5,6-dihydro-4*H*-1,3-thiazines **52a,b** (57% and 75%) were prepared via the treatment of the *N*-(2-hydroxyethyl)propionamide **53** with the Lawesson’s reagent followed by exposure to a solution of K_2_CO_3_ (Scheme 13) [35,36].

##### Synthesis of 2-imino-1,3-thiazinane Derivatives

Chemoselective synthesis of ferrocene-containing 1,3-thiazinan-2-imines **58a**–**m** via the reaction between 3-aryl-amino-1-ferrocenylpropan-1-ols **55a**–**m** and phenyl isothiocyanate in acidic medium. The intermediate *β*-hydroxy thioureas **56** were generated in situ using ultrasound irradiation and the cyclizations were achieved by the addition of acetic acid to give the corresponding 3-aryl-6-ferrocenyl-*N*-phenyl-1,3-thiazinan-2-imines **58a**–**m** (yields 52–90%) instead of 3-arylamino-1-ferrocenylpropan-1-ols **57** (Scheme 14) [37].

The mechanism for the formation of ferrocenyl 1,3-thiazinane-2-imine **58a**–**m** was illustrated in Scheme 15. Thiourea derivatives **56** were via nucleophilic attack of the amine **55** on the isothiocyanate. Under acidic conditions, the thiourea cyclized via the thione-group with the elimination of H_2_O molecule to give intermediate **61** through intermediates **59** and **60,** respectively. Intermediate **61** was deprotonated to give **58** (Scheme 15).

##### Synthesis of 1,3-thiazinane-4-one Derivatives

A variety of methods were made to synthesize 2-imino-1,3-thiazinane, based on the cyclization of acyl thioureas containing an *α*,*β*-unsaturated acid fragment. Reactions of acryloyl chloride with thiourea or with *N*-substituted thioureas, no *N*-acryloylthioureas **62** were isolated and hydrochlorides of 3-substituted-1,3-thiazinane-4-one **63a**–**d** were obtained. 2-Imino-1,3-thiazinane -4-one **65a,b** with a substituent on the exocyclic *N*-atom, were synthesized via thermal cyclization of methacryloyl thioureas **64a,b** (Scheme 16) [38].

The syntheses of 3-unsubstituted 2-imino-1,3-thiazinan-4-ones **68** and **69**, were based on the reaction of *α*,*β*-unsaturated carboxylic esters **66** with thioureas, including isolation and subsequent cyclization of hydrochlorides or sulfates **67** in the presence of aqueous ammonia or sodium acetate [39]. In the case of maleic or fumaric acids, hydrochlorides of 2-imino-thiazinans **69** were obtained in one-pot synthesis (Scheme 17) [40].

6-Unsubstituted 2-imines-1,3-thiazinane-4-one **72,** were synthesized via reaction of *β*-propiolactone **70** [41,42] and its derivatives with thioureas. At the first step, acids **71** were isolated; the cyclization of **71** in acetic anhydride or its mixture with pyridine gave thiazinan-4-ones **72.** Thiosemicarbazones reacted similarly to give 1,3-thiazinan-4-ones ((*E*)-2-((*E*)-((5-nitrofuran-2-yl) methylene)hydrazono)-1,3-thiazinan-4-one) (**73**) (51%) in one pot procedure [43] (Scheme 18).

Thiazinanones **76a**–**n** were synthesized via three-component reactions between aldehydes, 2-morpholinoethanamine (**74**) and 3-mercaptopropionic acid under both thermal and ultrasonication conditions. The products were formed via the intermediates **75a**–**n** [44] (Scheme 19).

*(Z)*-2-[(2,4-Dimethylphenyl)imino]-1,3-thiazinan-4-one **78** was prepared according to the procedure reported by Mansuroğlu et al.[45]. 3-Chloropropionyl chloride was reacted with potassium thiocyanate and 2,4-dimethylaniline, after acidification *N*-(3-chloropropionyl)-*N’*- (2,4-di-methylphenyl)thiourea (**77**) was formed. The substituted thiourea **77** was refluxed in toluene/acetone media to afford *(Z)*-2-[(2,4-dimethyl-phenyl)imino]-1,3-thiazinan-4-one (**78**) (Scheme 20) [46].

3-Mercaptopropionic acid reacted with ammonia or primary amines and aryl aldehydes to give 2- and 2,3-substituted-1,3-thiazinan-4-ones **79a**–**s**. The corresponding 1,3-thiazinan-4-one- 1,1-dioxide derivatives **80** (27–95%) were obtained from the synthesized substituted 1,3-thiazinan-4-ones **79** (11–74%) via oxidation using KMnO_4_ (Scheme 21) [6].

The reaction of keto fatty acids and long-chain aldehydes with 3-mercapto-propionic acid in the presence of ammonium carbonate resulted in the formation of thiazanone derivatives. The treatment of methyl 10-oxoundecanoate **81a**, methyl 9-oxostearate **81b** and octadecanal **81c** with 3-mercaptopropionic acid in the presence of ammonium carbonate ((NH_4_)_2_CO_3_) the thiazanone derivatives were obtained, 9-(2-methyl-4-oxo-1,3-thiazinan-2-yl)nonanoic acid **82a**, 8-(2-nonyl-4-oxo-1,3-thiazinan-2-yl)octanoic acid **82b** and 2-heptadecyl-1,3-thiazinan-4-one **82c**, respectively. Under the same conditions thiazanones **84a,b** and **85a,b** were obtained from the *vicinal*-dioxo ester **83a,b** (Scheme 22) [47].

Azeotropic reflux of *(E)*-methyl 4-oxo-octadec-2-enoate (**86**) [48] with methyl 3-mercaptopropionate and ammonium carbonate afforded the thiazinane, as a mixture of isomers **87** and **88** (Scheme 23) [49].

Dialkyl phosphites **89** reacted with difluoro- or trifluoroacetonitriles in the presence of a catalytic amount of nitrogen base to form iminophosphonates **90** and **91** as diastereoisomers. Cyclo-condensation of iminophosphonates **90** and **91** with 3-mercaptopropionic acid furnished 1,3-thiazinan-4-ones **92a**–**c** in good yields 79–88% (Scheme 24) [50].

Three-component reactions between amines or amino acids, aldehydes and 3-mercaptopropionic acid were catalyzed dicyclohexylcarbodimide (DCC) afforded metathiazanones **93a**–**d** in yields 51–92% (Scheme 25) [51].

The ring enlargement of 2,3-diphenylcyclopropenone **94** using 1-amino-2-substituted alkene- 1-thiols **95a**–**c** afforded different 5,6-diphenyl-2-(substituted-2-ylidene)-1,3-thiazinan-4-one **96a**–**c** (68–93%) (Scheme 26) [52].

Coupling of bis-5,5′-methylenebis(2-hydroxybenzaldehyde) (**97**) with bromo-acetaldehyde diethyl ether furnished the desired diacetal (5,5′-methylenebis(2-(2,2-diethoxyethoxy)benzaldehyde)) (**98**) in 74% yield. Deacetylation the diacetal **98** followed by intramolecular aldol condensation and acid-catalyzed dehydration afforded benzofuran-2-al dimer (5,5′-methylenebis(benzofuran-2- carbaldehyde)) **99** (88%). Condensation of **99** (in excess) with alkyl-, cycloalkyl-, aryl- and aralkyl amines gave bis-imines **100a**–**j** (83–94%). Subsequent cyclization of bis-imines **100a**–**j** through condensation with 3-mercaptopropionic acid furnished bis-(benzofurane-1,3-thiazinan-4-one) derivatives **101a**–**j** (Scheme 27) [53].

*N*^1^-(7-Chloroquinolin-4-yl)alkane diamines **102a**–**c** reacted with aldehydes in THF under ice-cold conditions, followed by addition of 3-mercaptopropanoic acid in presence of dicyclohexylcarbodimide (DCC) or in toluene under reflux afforded 2-(alkyl/aryl)-3-(2-((7-chloro quinolin-4-yl)amino)ethyl)-1,3-thiazinan-4-one derivatives **103a**–**i** in 48–67% yields (Scheme 28) [54].

The reaction of 2,4-dichlorobenzoic acid with *p*-methoxyaniline gave diphenylamine **104** on treatment with POCl_3_ cyclized to 6,9-dichloro-2-methoxyacridine (**105**) (85%). The acridine **105** reacted with 1,3-propandiamine afforded *N*^1^-(6-chloro-2-methoxyacridin-9-yl)propane-1,3-diamine (**106**). Compound **106** reacted with aldehydes and 3-mercaptopropionic acid in the presence of dicyclohexylcarbodimide (DCC) as a dehydrating agent furnished quinacrine[1,3]-thiazinan-4-one derivatives **107** in yields 60–78% (Scheme 29) [55].

3-Alkyl-2-aryl-1,3-thiazinan-4-one derivatives **109a**–**c** were synthesized via the routes outlined in Scheme 30. Treatment of amines with 4-methylthiobenz-aldehyde and thioglycolic acid in dry toluene in the presence of *p*-TsOH under reflux afforded 3-alkyl-2-(4-methylthiophenyl)- 1,3-thiazinan-4-one (**108**). Oxidation of **108** using 30% H_2_O_2_ in methanol in the presence of trace amount of tungsten oxide (WO_3_) gave 3-alkyl-2-(4-methylsulfonylphenyl)-1,3-thiazinan-4-one **109a**–**c** (35–75%). For low boiling point amines, the intermediate imine products **110**-were obtained by the reaction with 4-methylthiobenzaldehyde in anhydrous DMF. Subsequent oxidation **110** with hydrogen peroxide and WO_3_ in methanol solution afforded the (*E*)-*N*-(4-(methylsulfonyl benzylidene)alkyl-1-amine **111**. Reaction of **111** with mercaptopropionic acid under reflux gave **109d**–**f** (12–45%) (Scheme 30) [56].

3-Hydroxy-*N*-(4-oxo-2-phenyl-1,3-thiazinan-3-yl)-8-(trifluoromethyl)quino-line-2-carboxamide derivatives **113a**–**j** were synthesized by one-pot three component cyclocondensation reaction between quinoline hydrazide **112**, substituted benzaldehyde and 3-mercaptopropionic acid in the presence of 1-ethyl-3-(3-dimethylamino-propyl)carbodiimide EDC (Scheme 31) [57].

Hydrazinecarboxamides **114a**–**d** reacted with 3-mercaptopropionic acid in presence of SiCl_4_ gave 1,3-thiazinan-4-one as urea derivatives **115a**–**d** (39–43%), (Scheme 32) [58].

Hassan et al. reported that diastereoselective reaction between 4-substituted 1-(2,4-dinitrophenyl)thiosemicarbazides **116a**–**e** and 2,3-diphenylcycloprop-2-enone **94** under refluxing ethanol furnished racemic 2-(2,4-dinitrophenyl)hydrazono)-5,6-diphenyl- 1,3-thiazinan-4-ones **117a**–**e** (79–83%) as a major product and (*Z*)-*N‘*-(2,4-dinitrophenyl)- 2,3-diphenylacrylo hydrazide **118** (8–12%) as minor product (Scheme 33) [59].

The mechanism for the formation of products **117a**–**e** is presented in Scheme 34. The sulfur atom attacks the conjugate double bond of **94** forming the intermediate **119**. The intermediate **119** underwent ring opening to compound **120**. Intramolecular nucleophilic attack of N-4 on C=O afforded the intermediate **121** which rearranged to give **117a**–**e**. On the other hand, N-4 attacks the carbonyl group of **94** with the formation of **117a**–**e** via intermediates **122** and **123** (Scheme 34).

One-pot, three-component reactions of fluoro substituted benzaldehydes **124a,b** with amines and mecaptopropanoic acid afforded 1,3-thiazinan-4-one **125a,b**. Under microwave-assisted palladium-catalyzed coupling reactions in presence of boronic acid, thiazinanone **125a,b** gave the biaryl thiazinanones **126** and thioarylthiazinanones **127**. The microwave-assisted reactions were carried out using Pd(dppf)Cl_2_ [(1,1′-bis(diphenylphosphino)ferrocene) dichloro palladium(II)] as a catalyst, K_2_CO_3_ as a base and 4:4:1 acetone/toluene/water as a co-solvent (Scheme 35) [60].

Polyfluoroalkanethioamides using BF_3_ in diethyl ether and ethyl acrylate were reacted and afforded 1,3-thiazinan-4-one **130a**–**c** (25–50%) through the formation of intermediates **128** and **129** (Scheme 36) [61].

4-Oxo-1,3-thiazinan-11-oxoundecensulfanyl propanoic acid **134** was prepared in two steps: The hydrazine (*N*’-(3-nitrobenzylidene)undec-10-enehydrazide) (**132**) was first prepared by refluxing 10-undecenoic acid hydrazide **131** with *m*-nitrobenzaldehyde in anhydrous benzene. The compound **132** was then reacted with 3-mercaptopropionic acid, uncyclized adduct **133** (58%) was formed as aside product along with 4-oxo-1,3-thiazinan-11-oxoundecyl thiopropanoic acid **134** (26%) (Scheme 37) [62].

Isonicotinohydrazide (**135**) was reacted with aldehydes and 3-mercaptopropionic acid in presence of 1-ethyl-3-(3-dimethylaminopropyl)-carbodiimide (EDC) gave 1,3-(thiazinan-3-yl)- isonicotinamides **136a**–**f**’ in moderate to high yields 60–93% (Scheme 38) [63].

Similarly 4-(2-(methyl(pyridin-2-yl)amino)ethoxy)benzaldehyde (**137**) was reacted with appropriate primary amines (RNH_2_) and 3-mercaptopropinoic acid in presence of 1-ethyl-3-(3-dimethylaminopropyl)carbodiimide (EDC) at room temperature to give thiazinan-4-ones **138a**–**f** (25–54%) (Scheme 39) [64].

On the other hand, 4,5-dibromo-1-methyl-*N*-(4-oxo-2-aryl-1,3-thiazinan-3-yl)-1*H*-pyrrole-2-carboxamide **140a**–**h** were synthesized in a quantitative yields via one-pot three component condensation between 4,5-dibromo-1-methyl-1*H*-pyrrole-2-carbohydrazide (**139**), aromatic aldehydes and 3-mercaptopropionic acid in the ratio 1:2:3 in the presence of 1-ethyl-3-(3-dimethylaminopropyl) carbodiimide (EDC) (Scheme 40) [65].

4-(4-Oxo-6-phenyl-1,3-thiazinan-2-ylideneamino)benzoic acid (**142**) were obtained during the stirring of (*E*)-4-(3-cinnamoylthioureido) benzoic acid (**141**) with sodium ethoxide at room temperature, then the reaction mixture was neutralized by HCl (Scheme 41) [66].

The base-catalyzed reactions of *β*-oxonitriles (ethyl 2-cyanoacetate) with ethyl 3-mercaptopropanoate were illustrated in Scheme 42. The more reactive *β*-oxonitrile reacted with *β*-mercaptoester afforded ethyl (*E*)-4-oxo-[1,3]thiazinan-2-ylidene)ethanoate (**144**), after the cyclization of the intermediate ethyl (*Z*)-3-amino-3-(2-ethoxycarbonylethylsulfanyl)propenoate (**143**) (Scheme 42) [67].

The reaction of acyl thiourea **145** with potassium thiocyanate via an unusual thiocyanic acid elimination through the formation of intermediates **146**–**148** afforded 2-imino-3-phenyl -1,3-thiazinan-4-one **(149)** (Scheme 43) [68].

Allylic bromides **151a**–**h** were prepared from **150a**–**h** and reacted with thiourea in a 3:1 mixture of acetone:water at room temperature then reacted with an aqueous base of isothiouronium salts **152** gave 2-amino-1,3-thiazin-4-ones **153** as insoluble solids [69,70]. Transformation of 2-aminothiazin-4-one **153a** into thiazinane-2,4-dione **158a** was achieved by hydrolysis in an acidic medium [71]. The development of this method was achieved via a two-step (one-pot) method, initially: acetylation of 2-amino-thiazinan-4-one **153a** followed by mild hydrolysis of the acetylated intermediates. 2-Iminothiazinan-4-one **153a** was acetylated using acetic anhydride to give an approximately 1:1 mixture of two (out of four) possible acetylated isomers **154**–**157.** Acetylation/hydrolysis protocol was then extended to other thiazine-4-ones **153** with the formation of the expected 1,3-thiazinane-2,4-diones **158a**–**h** (58–85%) (Scheme 44) [72].

The condensation of 4-(3-isopropyl-4-methoxyphenoxy)-3,5-dimethylbenz-aldehyde (**159**) with thiazolidine-2,4-dione (**160**) under basic conditions gave the rearranged thiazinane-2,4-dione **162** (12%) in addition to thiazolidine-2,4-dione (**161**) (62%) (Scheme 45) [73].

Under ice-condition reactions of oxazolidinethiones **163** with 3-bromo-propionyl chloride in methylene chloride gave 1,3-thiazinane-2,4-diones **164a**–**d** (23–78%) (Scheme 46) [74].

The formation of 1,3-thiazinane-2,4-diones **164a**–**d** took place through the formation of both, the bromoamide **166** and S-alkylated intermediate **167** via *N*-acylation or intramolecular substitution reaction, respectively. Both intermediates **166** and **167** gave the immonium salts **168**, which lost HX molecule with ring-opening to give **164a**–**d (**Scheme 47).

##### Synthesis of 1,3-thiazinane-2-thione-4-one Derivatives

Arylideneoxazalones **170a**–**g** were added to (4-oxobutyl)carbamodithioic acid (**169**) and the mixture was subjected to microwave irradiation in presence of montmorillonite K10 (SiO_2_/Al_2_O_3_), basic and neutral alumina and silica gel-forming Michael adducts **171** which were cyclized to 1,3-thiazinane derivatives **172a**–**g** in yields 76–91% (Scheme 48) [75].

Pseudo-peptide containing 4-oxo-2-thioxo-1,3-thiazinane **175** in 65% yield, was obtained via Isocyanide-based six-component reactions with itaconic anhydride **173** (Scheme 49) [76].

Initially, carbamodithioic acid was formed from a primary amine and carbon disulfide. Then, Michael addition of carbamodithioic acid (in situ prepared) to itaconic anhydride **173** afforded the intermediate **176**, which underwent an intramolecular cyclization to give **174**. The addition of the carbenoid-C atom of the isocyanides onto the iminium group followed by the addition of the carboxylate ion onto the C -atom of the nitrilium ion leads to the formation of the adduct **178**, which underwent intramolecular acylation (Mumm rearrangement) [77] to give **175** (Scheme 50).

Similarly, one-pot three-component reaction of primary amines (RNH_2_), carbon disulfide (CS_2_) and itaconic anhydride (**173**) in water resulted in the formation of 2-(3-alkyl-4-oxo-2-thioxo- 1,3-thiazinan-5-yl) acetic acid derivatives **174a**–**i** in 68–95% yields (Scheme 51) [78].

Carbamodithioic acid was formed from a primary amine and carbon disulfide. Then it underwent Michael addition to itaconic anhydride **173** to give intermediate **176**, which underwent an intramolecular cyclization to afforded **174** (Scheme 52).

On the other hand, the one-pot reaction between primary amines and carbon disulfide in the presence of acryloyl chloride afforded 2-thioxo-1,3-thiazinane-4-one derivatives **179** in 70–89% yields (Scheme 53) [79].

##### Synthesis of 1,3-thiazinane-2-thione Derivatives

Allylamines **181**, which easily, obtained via the reaction of acetates **180** of Baylis–Hillman alcohols with appropriate primary amines. The allylamine **181** was transformed into *cis*-5,6-disubstituted-1,3-thiazinane-2-thione derivatives **182** (82–94%) via the reaction with carbon disulfide in the presence of dimethylaminopyridine (DMAP) (Scheme 54) [80].

A plausible mechanism for the formation of thiazinanes **182a**–**m**, with 5,6-*cis*-stereochemistry, is presented in Scheme 55. Nucleophilic attack of amine **181** onto CS_2_ gave the intermediate thiocarbamate ion (S=C–S^−^) **183**, which underwent Michael addition to the *α,β*-unsaturated nitrile moiety to give the carbanion **184**. Then protonation of the carbanion species **184** from the less hindered side, gave the thiazinanes **182** with 5,6-cis-stereoselectivity.

Solvent-free one-pot stereoselective synthesis of 1,3-thiazinane-2-thione derivatives **186** (85–89%) was achieved through the interaction between primary amines, carbon disulfide and *α*,*β*-unsaturated aldehydes (Scheme 56) [81].

The 1,3-thiazinane-2-thione **186** was formed upon the nucleophilic addition of the amine to carbon disulfide (S=C=S) and formation of dithiocarbamate, followed by addition to the *α,β*–unsaturated aldehyde to form intermediate **188**, which underwent intramolecular nucleophilic cyclization on the carbonyl group to afforded thiazinane-2-thione **186** (Scheme 57).

*N*-Alkyl-1,3-thiazine-2-thiones **188** was prepared from the reaction of 3-bromopropylamines [82] or substituted thiourea via iminothiazines **189** [83] with carbon disulfide. In addition, it was obtained from dithiocarbamic acids with 1,3-dibromopropane [84] (Scheme 58) [85].

1,3-Thiazinane-2-thione **188** (50% yield) was prepared via the treatment of 3-aminopropan-1-ol with sulfochloridic acid followed by carbon disulfide (CS_2_) (Scheme 59) [86].

Dithiocarbamates were reacted with 1,3-dibromopropane in basic medium gave 3-bromopropyl alk/arylcarbamodithioate **189**, which cyclized to both 1,3-thiazinan-2-thione derivatives **188a**–**h** (30–73%) and 2-imino-1,3-dithian derivatives **190a**–**h** (5–34%) (Scheme 60) [85].

2-Oxo-thiophen acetamide **192** was reacted with aryl isothiocyanates **191a**–**c** yielding butyric acid derivatives **193a**–**c**. Cyclization of **193a,b** in the presence of dicyclohexylcarbodimide (DCC) and 4-pyrrolidinopyridine yielded 1,3-thiazipane derivatives **194**, which underwent ring transformation to afford 1,3-thiazinan-2-thione derivatives **196a,b** (78% and 54%) **(**Scheme 61) [87].

##### Synthesis of 1,3-thiazinane-4-carboxylic Acid Derivatives

The well-known cyclization reactions of β/γ-aminoalkylthiols (containing both SH and NH_2_ groups) with organic aldehydes, which form the thiazolidine and thiazinane derivatives, have been widely used to design fluorescent probes for the detection of the concentration of Cys and HCys in living tissues.

DL-Homocysteine was reacted with benzaldehyde in absolute ethanol for three days, afforded the stereoisomers (2*S*,4*R*)-, (2*S*,4*S*)-, (2*R*,4*R*)-, (2*R*,4*S*)-2-phenyl-1,3-thiazinane-4-carboxylic acid (**197**) (Scheme 62) [88].

The reaction of Ir(pba)_2_(acac) **198** (Hpba = 4-(2-pyridyl)benzaldehyde; acac = acetylacetone) with homocysteine under stirring for 12 days in mixture of CH_2_Cl_2_/MeOH as solvent (2:1 *v/v*) afforded Iridium complex of thiazinane **199** (16%) (Scheme 63) [89].

Homocysteine was reacted with 2′-((2-hydroxyethyl)amino)-[1,1′:4′,1″-terphenyl]-4,4″-dicarb-aldehyde **200** bearing electron-donating group (-NH(CH_2_)_2_OH) and electron withdrawing group (-CHO) gave 2,2′-(2′-((2-hydroxyethyl)amino)-[1,1′:4′,1″-terphenyl]-4,4″-diyl)bis(1,3-thiazinane-4-carboxylic acid) **201** (Scheme 64) [90].

4-(6,11-Dioxo-6,11-dihydro-1*H*-anthra[1,2-*d*] imidazol-2-yl)benzaldehyde (**203**) was synthesized through condensation between 1,2-diaminoanthraquinone (**202**) and terephthalaldehyde. In addition, imidazophenanthrolin benzaldehyde **206** was obtained by refluxing a mixture of 1,10-phen anthroline-5,6-dione **205** and terephthalaldehyde. The two ligands **202** and **206** were cyclized with homocysteine furnished anthra[1,2-d]imidazolyl-1,3-thiazinane-4-carboxylic acid **204** and imidazophenanthrolin-1,3-thiazinane-4-carboxylic acid **207**, respectively (Scheme 65) [91].

Ruthenium (II( complexes containing aldehyde groups **208** were characterized to recognize homocysteine via the formation of thiazinane **209**. A strong luminescence response was found upon the reaction of the ruthenium (II) chromophore **208** with homocysteine (Scheme 66) [92].

Tetraphenylethylenedialdehyde (**210**) was used for the detection of homo-cysteine via the formation of ((*E*)-2,2′-((1,2-diphenylethene-1,2-diyl)-bis(4,1-phenylene))bis(1,3-thiazinane-4- carboxylic acid)) **211** in DMSO under buffering conditions (pH = 7.4) as shown in Scheme 67 [93].

The trialdehyde **212** showed high selectivity for homocysteine at pH = 6.0 via the formation of thiazinane **213** (Scheme 68) [94].

Suzuki–Miyaura–cross-coupling [95] of 4-chloro-7-nitrobenzo[1,2,5]-oxadiazole **214** with 4-formylphenylboronic acid **215** yielded (4-(7-nitrobenzo-[*c*][1,2,5]oxadiazol-4-yl)benzaldehyde) (**216**), which was reacted with homocysteine afforded a highly fluorescent compound oxadiazolyl-1,3-thiazinane-4-carboxylic acid **217** through the cyclization with the aldehydic group (Scheme 69) [96].

The azo dyes 4-[[40-(*N,N*-dimethylamino)phenyl-10-]azo]benzaldehyde **218** and 4-[[4′-(*bis*(2-hydroxyethyl)amino)phenyl-10-]azo]-3-nitrobenzaldehyde **220** were reacted with cysteine and homocysteine. The reaction of **218** and **220** with homocysteine afforded very stable derivatives thiazinane **219** and **221** under neutral pH conditions (Scheme 70) [97].

Quinoline derivative **222** was also used to detect homocysteine depending on, the formation of thiazinane **223** ring through cyclization reaction (Scheme 71) [98].

1,10-Bi-2-naphthol **224** based dialdehyde was found to exhibit selective fluorescent response towards cellular thiols, cysteine and homocysteine. 2,2′-Dihydroxy-[1,1′-binaphthalene]-3,3′-dicarb aldehyde (**224**) reacted with homocysteine, resulted in the formation of thiazinane **225** (Scheme 72) [99].

The addition of homocysteine into 5-(benzothiazol-2-yl)-4-hydroxyiso-phthalaldehyde (**226**) *o*-aldehyde group was transformed into (2*S*,4*R*)-benzo[*d*]thiazol-1,3-thiazinane-4-carboxylic acid derivatives **227** (Scheme 73) [100].

6-((4-(Dimethylamino)phenyl)ethynyl)quinoline-2-carbaldehyde (**228**) showed high selectivity in the detection of cysteine and homocysteine, because of the formation of thiazolidine and thiazinane derivatives. The quinoline-2-carbaldehyde **228** was reacted with homocysteine afforded 2-(6-((4-(dimethylamino)-phenyl)ethynyl)quinolin-2-yl)-1,3-thiazinane-4-carboxylic acid **229** (Scheme 74) [101].

#### 2.1.3. Synthesis of 1,4-thiazinane Derivatives

##### From Diazabutadiene and Butylaminoethanethiol

Addition of 2-(butylamino)ethanethiol **231** to 1,2-diaza-1,3-butadiene **230** resulted in the formation of hydrazone 1,4-adduct intermediate **232**. The reaction between **231** and 1,2-diaza-1,3-butadienes **230** containing an ester group in position 4 of the heterodiene system gave 2-[1-(4-butyl-3-oxo-1,4-thiazinan-2-yliden)ethyl]-1-hydrazinecarboxylates **234a,b** (96% and 63%) via intermediates **232** and **233** (Scheme 75) [102].

##### From Cyclic Sulfamidates

Cyclic sulfates and cyclic sulfamidates represented a versatile class of functionalized and enantiomerically pure electrophiles. A six-ring *N*-heterocycle ((*S*)-4,5-dibenzyl-1,4-thiazinane-3-one) **237** (94%) was formed through a regioselective nucleophile displacement on **235** via reaction with methyl 2-mercaptoacetate and subsequent lactamization of (*S*)-benzyl(1-((2-methoxy-2-oxoethyl) thio)-3-phenylpropan-2-yl)sulfamate (**236**) (Scheme 76) [103].

##### From Diethyl 2,2-sulfonyldiacetate

Diethyl 3,5-diphenyl-1,4-thiazinane-2,6-dicarboxylate 1,1-dioxide **239a**–**m** (79–91%) were formed by reacting diethyl 2,2-sulfonyldiacetate (**238**) and aryl/heteroyl aldehydes in water, in the presence of ammonium acetate (Scheme 77) [104].

##### From Ethyl 2-[(2-oxo-2-arylethyl)sulfonyl]acetate

The reaction of sulfonylacetate **240**, aromatic aldehydes and amines in presence of L-proline (**241**) as green catalyst furnished 1,1-dioxo-1,4-thiazinane-2-carboxylates **242a**–**a’** (72–90%) (Scheme 78) [105].

L-Proline catalyzed the reaction between sulfonyl acetate and aromatic aldehyde via the formation of enamine-imine intermediates **243** and **244**, respectively, followed by dehydration of **244** to give intermediate **245**. Losing of, proline moiety of **245** via the attack of the amine result in the formation of intermediate **246**, which was condensed with another molecule of the aldehyde followed by intramolecular cyclization with deprotonation to furnish **242** (Scheme 79).

#### 2.1.4. Synthesis of Fused Thiazinane Derivatives

##### Synthesis of Tetrahydrocyclopenta[e][1,3]thiazinan-2,4-dione

Tetrahydrocyclopenta[e][1,3]thiazinan-2,4-dione **250** was formed by reacting 2-thio cyanatocycIopent-1-ene-1-carboxylic acid (**248**) and thionyl chloride at room temperature via ring closure of the intermediate carboxylic acid chloride **249** (Scheme 80) [106].

##### Synthesis of 1,3-benzothiazinan-4-one Derivatives

4-Methylsulfonylbenzaldehyde (**251**) was reacted with aromatic amines and thiosalicylic acid (**252**) in the presence of *p*-TsOH gave 2-(4-methylsulfonylphenyl)-3-substituted-1,3-benzothiazinan- 4-one **253a**–**f** (33–73/5) (Scheme 81) [107].

##### Synthesis of Tetrahydropyrido[2,1-b]-[1,3]thiazine-7-carboxylate

In multicomponent reactions, ethyl 6-amino-8-(4-methoxy phenyl)-9-nitro-2,3,4,8-tetrahydro- pyrido[2,1-*b*][1,3]thiazine-7-carboxylate **256** were synthesized. Initially, 3-aminopropanethiol was reacted with (2-nitroethene-1,1-diyl)bis-(methylsulfane) (**254**) in dry ethanol afforded 2-(nitro-methylene)-1,3-thiazinane (**255**). In the second step, compound **255** reacted with ethyl cyanoacetate and *p*-methoxybenzaldehyde furnished ethyl 6-amino-8-(4-methoxyphenyl)-9-nitro- 2,3,4,8-tetrahydropyrido[2,1-*b*][1,3]thiazine-7-carboxylate (**256**) (Scheme 82) [108].

##### Synthesis of [1,3]thiazino[3,2-a]indole

Thiazinane[3,2-a] indole **261** was synthesized from 1-(3-(acetylthio)propyl)-2,3,3-trimethyl-3*H*- indol-1-ium iodide (**258**). *N*-Substituted-3*H*-indoles (**258**) were obtained via nucleophilic substitution of 2,3,3-trimethyl-3*H*-indole (**257**) with alkyl halides (*S*-(3-iodopropyl)ethanethioate). Condensation of **258** with the reactive cyanine derivative ((*E*)-1-ethyl-3,3-dimethyl-2-(2-(*N*-phenylacetamido) vinyl)-3*H*-indol-1-ium chloride) (**259**) afforded protected 1-(3-(acetylthio)propyl)-2-((1*E*,3*E*)-3-(1-ethyl-3,3-dimethylindolin-2-ylidene)prop-1-en-1-yl)-3,3-dimethyl-3*H*-indol-1-ium iodide) (**260**). After deprotection under basic conditions [1,3]thiazino[3,2-*a*] indole **261** was obtained in high yield 93% (Scheme 83) [109].

##### Synthesis of [1,3]dioxolo[4′,5′:3,4]pyrido[2,1-b][1,3]thiazinanone

Microwave-assisted one-pot Staudinger/aza-Wittig/cyclization reaction using **262a** and **262b** as the starting materials afforded two diastereoisomers of the bi/tricyclic azasugars **263a,b** and **264** in satisfying yields with low stereoselectivity (in total yields 62%) (Scheme 84) except the case of the reaction of **262b** with 3-mercaptopropionic acid stereospecifically afforded a single diastereoisomer **(**(3a*S*,4*R*,5*R*,-10a*R*,10b*S*)-4-hydroxy-5-((trityloxy)methyl)hexahydro-[1,3]dioxolo-[4′,5′:3,4]pyrido-[2,1-*b*][1,3]-thiazin-7(3a*H*)-one) (**263b**, 71%), possibly due to the synergistic hindrance effects of the *cis* neighboring cyclic 2,3-isopropylidene and 5*β*-group in **262b**, which made a dominant *exo*-attack of the sulfur atom to the intermediate imine (Scheme 84) [110].

The aza-sugar **262** was cyclized to intermediate **265** in presence of PPh_3_ with losing of N_2_ and Ph_3_P=O. Nucleophilic attack of the thiol–group of the mercaptopropionic acid on the imine carbon of **265** gave intermediate **266**. Intramolecular cyclization of intermediate **266** afforded **263** (Scheme 85).

##### Synthesis of Octahydrobenzo[f][1,3]thiazino[2,3-b]quinazoline

Unsymmetrical quinazoline-3-thione (1-(4-chlorophenyl)-1,2,5,6-tetrahydro-benzo[*f*]quinaz oline-3(4*H*)-thione) **268** (78%) was obtained from one-pot condensation of 2-tetralone **267**, *p*-chlorobenzaldehyde and thiourea in acidic medium. Condensation of quinazoline-3-thione **268** with 3-chloropropionic acid and 1,3-dibromopropane furnished thiazinoquinazoline derivatives **269** and **271** in 60% and 52% yield, respectively, instead of their regioisomers **270** and **272** (Scheme 86) [111].

#### 2.1.5. Synthesis of Spirothiazinane Derivatives

Condensation of fluorinated indole-2,3-diones **273a** and 1-acetylindole-2,3-diones **273b** with fluorinated aniline afforded 3-arylimino-2*H*-indol-2-ones **274**, which, in situ, were cyclized with 3-mercaptopropanoic acid to afford the spiro compounds **275**. In a few cases, intermediates isatin-3-anil **274** were isolated (Scheme 87) [112,113,114]. In addition, it was reported that fluorinated spiro[indoline-3,2′-[1,3]thiazinane]-2,4′-diones **275**, were synthesized via one-step synthesis through the formation of Schiff’s bases followed by cyclization with 3-mercaptopropanoic acid, both thermally and under microwave irradiation. The reactions were studied under different reaction conditions. It was observed that the yield was improved when the reaction was carried out under microwave irradiation [115]. Furthermore, Dandia et al. reported, a one-pot solvent-free synthesis of spiro[indole-3,2-[1,3]thiazinane]-2,4-diones **275a** (4 min, 140 °C (85%); 6 min, 135 °C (93%)) [108] from the reaction of intermediate **274** with 3-mercaptopropionic acid (Scheme 87) [112,113,114,115,116,117].

### 2.2. Reactions of Thiazinanes

#### 2.2.1. Reactions of 1,2-thiazinanes

##### N-Arylation of 1,2-thiazinane

*N*-Arylation reaction of 1,2-thiazinane-1,1-dioxide **41c** using Cu_2_O and Cs_2_CO_3_ in water gave *N*-arylmethanesulfonamide (2-phenyl-1,2-thiazinane-1,1-dioxide) **276** (82%) (Scheme 88) [118].

2-(4-Bromobenzyl)-1,2-thiazinane-1,1-dioxide **276b** was prepared via direct sulfonamidation of (4-bromophenyl)methanol. The reaction between (4-bromophenyl)methanol and 1,2-thiazinane- 1,1-dioxide **41c** was carried out using 2,3,4,5-tetrafluorophenylboronic acid, oxalic acid dihydrate and HFIP (hexafluoroisopropanol)/nitromethane mixture to afford **276b** (Scheme 89) [119].

#### 2.2.2. Reactions of 1,3-thiazinanes

##### Ring-opening of *N*-substituted 1,3-thiazinanes and Synthesis of Thioesters

1,3-Thiazinane-4-ones **130a**–**c** relatively stable in aqueous alkaline medium and are easily hydrolyzed under acidic conditions. Treatment of **130a–c** with conc. HCl resulted in, the formation of thioester derivatives **277a**–**c**. The possible reaction mechanism includes the elimination of ethanol from **130a**–**c** catalyzed by HCl with its subsequent addition to **278** giving intermediate **279**. Hydrolysis of the latter led to acyclic imine **280**, which was converted into **277a**–**c** (63–82%) under acidic conditions (Scheme 90) [61].

##### N-Alkylation of 1,3-thiazinane-2-thione

1,3-Thiazinane-2-thione (**188**) reacted with 1,2-dichloro-4-(1-chloroethyl)-benzene in presence of sodium hydride afforded 3-substituted 1,3-thiazinane-2-thione **281 (**Scheme 91) [120].

Similarly, 1,3-thiazinane-2-thione (**188**) was reacted with 1-chloro-4-(1-chloroethyl)benzene in presence of K_2_CO_3_ afforded 3-[1-(4-chlorophenyl)ethyl]-1,3-thiazinane-2-thione **282** (Scheme 92) [121].

In addition, 1,3-thiazinane-2-thione **188** [82] was condensed with 1-(1-hydroxyalkyl) benzotriazoles in the presence of boron trifluoride gave 3-(1-benzotriazolylalkyl)thiazine-2-thiones **283**. Nucleophilic substitution of benzotriazolyl group in 3-(1-benzotriazolyl alkyl)-1,3- thiazinane-2-thiones **283** using thiol compounds and in the presence of ZnBr_2_, 3-[1-(substituted sulfanyl)alkyl]-1,3-thiazinane-2-thiones **284a**–**c** were formed (78; 77; and 79%). On the other hand, 1,3-thiazinane-2-thiones **283** reacted with triethyl phosphite catalyzed by ZnBr_2_ in CH_2_Cl_2_ under reflux furnished 1-(2-thioxo-1,3-thiazinan-3-yl)alkylphosphonates **285a,b** (72; 77%) (Scheme 93) [85].

##### Synthesis of Bis-pyrrol and Bis-pyrrolothiazole

(*Z*)-2,2-Dichloro-1-(2-ethylidene-1,3-thiazinan-3-yl)butan-1-one **54a** underwent stereoselective copper-catalyzed radical cyclization (RC) under reaction conditions: CuCl with TMEDA or PMDETA in MeCN to give a 9:1 mixture of dimers **286** and **287**, respectively. Only traces of thioester **289** were indicated with the absence of dimer **288** (Scheme 94) [122].

##### Synthesis of Maleic Anhydride

Radical cyclization of (*Z*)-3-(2,2-dichloropropanoyl)-2-pentadecylidene-1,3-thiazinane **54** giving, thioacetal **286** and disulfide **288** (Scheme 94) [37]. Acetal **286** was oxidized to disulfide **287** using KI in water via the liberation of iodine (I_2_) as illustrated in scheme b. *N*-(3-Hydroxypropyl)-undec-10-enamide was applied in the reaction yield enhancement, which subjected to radical cyclization, followed by hydrolysis to furnish maleic anhydride **291** (60%) (Scheme 95) [123].

The liberation of iodine catalyzed radical oxidation of *S,S*-acetal **286** by accelerating ring-opening of intermediate **292** to **293**. In presence of H_2_O as a nucleophile attacked the imine-carbon in **293** to give the hydroxylated intermediate **294**. The liberation of HI from **294** gave disulfide **287**, which hydrolyzed to maleic anhydride **291** (Scheme 96).

##### Synthesis of Bis-thiazinethioether

Wang et al. made a series of multithioether derivatives **295a**–**c** (27–72%) using the reaction of thiazinan-2-thiones (**188**) with alkyl dibromides. The synthesized compounds were tested for antitumor activity (Scheme 97) [124].

##### Synthesis of Benzo[4,5]thieno[3,2-b][1,5]thiazocin-6(3H)-one

8-Membered thiazepinone ring analog (10-(benzyloxy)-4,5,6a,11b-tetrahydro-2*H*-benzo[4,5]- thieno[3,2-*b*][1,5]thiazocin-6(3*H*)-one) (**299**) was prepared from with the synthesis of **296** and **297** via the ring opening of 1,3-thiazinane-2-thione (**188**). Aminothiol **296** and **297** was isolated as a thiol/disulfide mixture and used directly in the aforementioned cyclo-condensation-deprotection sequence with methyl 5-(benzyloxy)-3-chlorobenzo[*b*]thiophene-2-carboxylate (**298**) to provide the desired 8-membered thiazepinone analogs (10-(benzyloxy)-4,5-dihydro-2*H*-benzo[4,5]thieno- [3,2-*b*][1,5]thiazocin-6(3*H*)-one) **299** (42%) (Scheme 98) [125].

## 3. Conclusions

This review was focused on the chemistry of 1,2-, 1,3- and 1,4-thiazinane derivatives, synthesis and reactions such as arylation, alkylation, ring-opening and dimerizations were presented. Some mechanisms were illustrated to evaluate the reaction pathways for the formation of thiazinane derivatives as well as their interactions. In addition, abbrev was demonstrated about the structure, biologic activities and the commercial drugs containing thiazinane rings on their structure cores.

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
