# Peer review of "Chemistry of Substituted Thiazinanes and Their Derivatives"

_molecules, 2020, doi:10.3390/molecules25235610_

Round 1

Reviewer 1 Report

The manuscript entitled ‘

Chemistry of substituted thiazinanes and their derivatives

by Alaa A. Hassan, Stefan Bräse, Ashraf A. Aly, and Hendawy N. Tawfeek

reports an extensive review on thiazinanes and their derivatives, focusing particular attention on their synthesis, sometimes besides mechanistic information.

The topic is interesting and the manuscript appears well structured, however, many typos or errors denote poor care in its preparation.

For example:

page 4, rows 6-8 must be rewritten

page 4, rows 11, 12, 15, 17, 19: compound 9, 10, 12, 13 are single molecules, not plural

page 5, row 1: idem

page 7, row 2; probably is right 31 instead of 33

page 8, rows 1-2: check the grammar

Scheme 10: Tf, not TF

Scheme 12: in 48 and 49 the N-NH bond must be longer, to be more evidenced

page 10, rows 4 and 7: check the verb

Scheme 14: Fc = ferrocenyl

Scheme 17: not clear: compound 69 bears a R group, not indicated; moreover, the Ph in N-Ph of 69, from which precursor comes? Again: HSO4- add the sign for the charge.

page 12, row2: its, not their

Scheme 19: over the last arrow, (b) or (c)

Scheme 21: the correspondence between R1 and R2 in figures and in caption must be checked

page 14, row 7: 84a,b

Scheme 22: 81a-c

Scheme 16: add the reagent bromoacetaldehyde diethyl ether in reagents and conditions

page 16, row 4: 103a-i

Scheme 29: TFH?

Scheme 30: phenathyl?

page 19, rows 8,9: 4NH might be written N-4 and indicated in Scheme 34

Scheme 34: The arrow towards 122 is wrongly written

Scheme 35: for 125°,b, define also R1

page 22: 3-mercaptopropinoic, correct

page 23, rows 11-12: add the verb

page 25, 76-91%

Scheme 50: add nitro group in formula 177

Scheme 51: hexcyl?

Scheme 53: flourobenzyl?

Scheme 65: therephtaldehyde

Page 35: quinoline derivative 222

Page 38, row 2: is not an hydrolysis!

References: publication year and journal name abbreviations must be checked carefully

Moreover, along the text, many errors on verbs and plural/singular words must corrected.

Author Response

Dear Sir,

Many thanks for your message concerning the referee comments on our review. Please find enclosed an attached file containing our revised manuscript and all corrections have been highlighted with yellow colour.

Response to reviewer 1 

page 4, rows 6-8 must be rewritten

  • It has been rewritten.

page 4, rows 11, 12, 15, 17, 19: compound 9, 10, 12, 13 are single molecules, not plural

  • It has been corrected.

page 5, row 1: idem

  • It has been corrected.

page 7, row 2; probably is right 31 instead of 33

  • It has been corrected.

page 8, rows 1-2: check the grammar

  • The grammar has been checked.

Scheme 10: Tf, not TF

  • It has been corrected.

Scheme 12: in 48 and 49 the N-NH bond must be longer, to be more evidenced

  • The structure has been cleaned up.

page 10, rows 4 and 7: check the verb

  • It has been checked.

Scheme 14: Fc = ferrocenyl

  • It has been corrected.

Scheme 17: not clear: compound 69 bears a R group, not indicated; moreover, the Ph in N-Ph of 69, from which precursor comes? Again: HSO4add the sign for the charge.

  • The R group has been inserted, the Ph deleted and the sign charge has been added .

page 12, row2: its, not their

  • It has been changed.

Scheme 19: over the last arrow, (b) or (c)

  • It has been inserted.

Scheme 21: the correspondence between R1 and R2 in figures and in caption must be checked

  • It has been checked and corrected.

page 14, row 7: 84a,b

  • It has been corrected.

Scheme 22: 81a-c

  • It has been corrected.

Scheme 16: add the reagent bromoacetaldehyde diethyl ether in reagents and conditions.

  • Scheme 16 does not contain bromoacetaldehyde as a reagent.

page 16, row 4: 103a-i

  • It has been corrected.

Scheme 29: TFH?

  • It has been corrected to "THF"

Scheme 30: phenathyl?

  • It has been corrected to "phenethyl"

page 19, rows 8,9: 4NH might be written N-4 and indicated in Scheme 34

  • It has been corrected.

Scheme 34: The arrow towards 122 is wrongly written

  • The arrow has been redrawing.

Scheme 35: for 125a,b, define also R1

  • It has been corrected and R1 have been added.

page 22: 3-mercaptopropinoic, correct

  • It has been corrected.

page 23, rows 11-12: add the verb

  • It has been added.

page 25, 76-91%

  • It has been corrected.

Scheme 50: add nitro group in formula 177

  • It has been added.

Scheme 51: hexcyl?

  • It has been corrected to "hexyl"

Scheme 53: flourobenzyl?

  • It has been corrected to "fluorobenzyl"

Scheme 65: therephtaldehyde

  • I has been corrected to " terephthalaldehyde"

Page 35: quinoline derivative 222

  • It has been corrected as " Quinoline derivative 222"

Page 38, row 2: is not an hydrolysis!

  • It has been changed to "dehydration"

References: publication year and journal name abbreviations must be checked carefully

  • The references have been checked

Moreover, along the text, many errors on verbs and plural/singular words must be corrected.

  • It has been corrected.

Response to reviewer 2

This is a comprehensive review of the field of thiazinane and does bring together a lot of information in one place. It is definitely valuable to the field and a good read for anyone interested in this scaffold. However, before it can be published quite a lot of mainly small things need to be addressed. These come together to detract from what is overall a nice summary of the current state of the field.

- In this sense it would be nice to see that authors provide atleast a full paragraph (or more) to the conclusion section to highlight some of the key features summarised in this review. If this was done, the review would have much more impact.

  • A paragraph was added to the conclusion.

- The whole paper needs to be thoroughly proof read and corrected including for English and grammar.

  • The paper has been checked for grammar. 

- The abstract while mostly technically correct suffers from bad English and needs to be proof read eg. 'which used for AIDS treatment' should be 'which has been used for AIDS treatment' etc

  • The abstract has been changed..

Specifically on the subject of ref 5 (10.1073/pnas.0402357101), this compound has been demonstrated to an ability to be used as an AIDS treatment. This is not proven so far as I see no human patients were involved in this reference, so it has been shown to have anti-HIV activity by a mechanism that should work as an anti-AIDS. This needs to be changed in the abstract and text otherwise the reader could be mis-led. This also needs to be addressed in the main text as there is no evidence that there is any ‘current’ work on this series. That fact that that functionality was used in a medchem program is interesting anyway and does not need to be over sold.

  • It has been corrected.

Why is this the structure of 5-(1,1-Dioxido-1,2-thiazinan-2-yl)-N-(4-fluorobenzyl)-8-hydroxy-1,6-naphthyridine-7-carboxamide and 5-(1,1-Dioxido-1,2-thiazinan-2-yl)-N-(4-fluorobenzyl)-8-hydroxy-1,6-naphthyridine-7-carboxamide not present in this introduction despite been mentioned in the abstract and main text.

  • The structure has been inserted in figure 3 with its name.

More details should be provided in the introduction to provide more context to the use of the thiazinane

  • Some details have been inserted and highlighted with yellow.

- Consistency and Errors

There are many small errors contained in this review eg figure 2

eg bretschneiderazines A not fully highlighting the 3-thiazine functionality as blue. This is a consistent error where in some cases not enough is highlighted an later on eg Scheme 87 the whole oxindole is highlighted aswell (stick to just the functionality)

  • Corrections have been made, and only thiazinane ring highlighted blue along the paper.

There is a lack of consistency of functional group representations nitro and aldehyde eg figure 4 vs Scheme 74 (aldehyde) and figure 4 vs Scheme 37 (nitro)

  • The nitro as well as the aldehydic groups have been changed .

This happens later in the text with SO2 vs O=S=O, there needs to be consistency. I personally prefer the correct angled O=S=O 

  • It has been changed.

- Examples of smaller errors - there are more not highlighted here

Figure 4 should have the names with a capital letter or not.

  • It has been corrected.

Scheme 5 - remove 'stirring' and throughout text unless really required eg Scheme 6, 61, 67 etc

  • It has been removed.

end of page 6 'OTF' should be 'OTf'

  • It has been corrected.

check scheme 12 for the correct representation

  • It has been checked and corrected.

Scheme 25 needs to show all substrates on scheme

  • The substrates have been inserted on the scheme.

Scheme 29 needs cleaning up, move CO2H to the lower position and clean up sizes

  • The structure has been cleaned up.

Scheme 35 n-butyl - 'n' should be in italics

  • It has been corrected

132 and 133 should be cleaned up and have an accurate representation

  • They have been cleaned.

Mumm rearrangement needs to cite - Mumm, O. Ber. Dtsch. Chem. Ges. 1910, 43, 886-893

  • It has been inserted as reference [77]

Scheme 53. issue with capital consistency

  • It has been checked and corrected.

4-[[40-(bis(2-hydroxyethyl)amino)phenyl-10-]azo]-3-nitrobenzaldehyde 'bis' should be italics

  • It has been corrected.

Scheme 75 and Scheme 97 need to be cleaned up, errors and alignment

  • They have been cleaned up and readjusted .

Scheme 98 is not aligned correctly and properly spaced out.

  • It has been readjusted.

 Sincerely yours

Prof. Dr. Alaa A. Hassan

Reviewer 2 Report

This is a comprehensive review of the field of thiazinane and does bring together a lot of information in one place. It is definitely valuable to the field and a good read for anyone interested in this scaffold. However, before it can be published quite a lot of mainly small things need to be addressed. These come together to detract from what is overall a nice summary of the current state of the field.

- In this sense it would be nice to see that authors provide atleast a full paragraph (or more) to the conclusion section to highlight some of the key features summarised in this review. If this was done, the review would have much more impact.

- The whole paper needs to be thoroughly proof read and corrected including for English and grammar.

- The abstract while mostly technically correct suffers from bad English and needs to be proof read eg. 'which used for AIDS treatment' should be 'which has been used for AIDS treatment' etc

Specifically on the subject of ref 5 (10.1073/pnas.0402357101), this compound has been demonstrated to an ability to be used as an AIDS treatment. This is not proven so far as I see no human patients were involved in this reference, so it has been shown to have anti-HIV activity by a mechanism that should work as an anti-AIDS. This needs to be changed in the abstract and text otherwise the reader could be mis-led. This also needs to be addressed in the main text as there is no evidence that there is any ‘current’ work on this series. That fact that that functionality was used in a medchem program is interesting anyway and does not need to be over sold.

Why is this the structure of 5-(1,1-Dioxido-1,2-thiazinan-2-yl)-N-(4-fluorobenzyl)-8-hydroxy-1,6-naphthyridine-7-carboxamide and 5-(1,1-Dioxido-1,2-thiazinan-2-yl)-N-(4-fluorobenzyl)-8-hydroxy-1,6-naphthyridine-7-carboxamide not present in this introduction despite been mentioned in the abstract and main text.

More details should be provided in the introduction to provide more context to the use of the thiazinane

- Consistency and Errors

There are many small errors contained in this review eg figure 2

eg bretschneiderazines A not fully highlighting the 3-thiazine functionality as blue. This is a consistent error where in some cases not enough is highlighted an later on eg Scheme 87 the whole oxindole is highlighted aswell (stick to just the functionality)

There is a lack of consistency of functional group representations nitro and aldehyde eg figure 4 vs Scheme 74 (aldehyde) and figure 4 vs Scheme 37 (nitro)

This happens later in the text with SO2 vs O=S=O, there needs to be consistency. I personally prefer the correct angled O=S=O

- Examples of smaller errors - there are more not highlighted here

Figure 4 should have the names with a capital letter or not.

Scheme 5 - remove 'stirring' and throughout text unless really required eg Scheme 6, 61, 67 etc

end of page 6 'OTF' should be 'OTf'

check scheme 12 for the correct representation

Scheme 25 needs to show all substrates on scheme

Scheme 29 needs cleaning up, move CO2H to the lower position and clean up sizes

Scheme 35 n-butyl - 'n' should be in italics

132 and 133 should be cleaned up and have an accurate representation

Mumm rearrangement needs to cite - Mumm, O. Ber. Dtsch. Chem. Ges. 1910, 43, 886-893

Scheme 53. issue with capital consistency

4-[[40-(bis(2-hydroxyethyl)amino)phenyl-10-]azo]-3-nitrobenzaldehyde 'bis' should be italics

Scheme 75 and Scheme 97 need to be cleaned up, errors and alignment

Scheme 98 is not aligned correctly and properly spaced out.

Author Response

(The authors gave the same response as above.)

Round 2

Reviewer 1 Report

The manuscript, after the revision, is suitable for publication

Author Response

Response to reviewer 1 

page 4, rows 6-8 must be rewritten

  • It has been rewritten.

page 4, rows 11, 12, 15, 17, 19: compound 9, 10, 12, 13 are single molecules, not plural

  • It has been corrected.

page 5, row 1: idem

  • It has been corrected.

page 7, row 2; probably is right 31 instead of 33

  • It has been corrected.

page 8, rows 1-2: check the grammar

  • The grammar has been checked.

Scheme 10: Tf, not TF

  • It has been corrected.

Scheme 12: in 48 and 49 the N-NH bond must be longer, to be more evidenced

  • The structure has been cleaned up.

page 10, rows 4 and 7: check the verb

  • It has been checked.

Scheme 14: Fc = ferrocenyl

  • It has been corrected.

Scheme 17: not clear: compound 69 bears a R group, not indicated; moreover, the Ph in N-Ph of 69, from which precursor comes? Again: HSO4add the sign for the charge.

  • The R group has been inserted, the Ph deleted and the sign charge has been added .

page 12, row2: its, not their

  • It has been changed.

Scheme 19: over the last arrow, (b) or (c)

  • It has been inserted.

Scheme 21: the correspondence between R1 and R2 in figures and in caption must be checked

  • It has been checked and corrected.

page 14, row 7: 84a,b

  • It has been corrected.

Scheme 22: 81a-c

  • It has been corrected.

Scheme 16: add the reagent bromoacetaldehyde diethyl ether in reagents and conditions.

  • Scheme 16 does not contain bromoacetaldehyde as a reagent.

page 16, row 4: 103a-i

  • It has been corrected.

Scheme 29: TFH?

  • It has been corrected to "THF"

Scheme 30: phenathyl?

  • It has been corrected to "phenethyl"

page 19, rows 8,9: 4NH might be written N-4 and indicated in Scheme 34

  • It has been corrected.

Scheme 34: The arrow towards 122 is wrongly written

  • The arrow has been redrawing.

Scheme 35: for 125a,b, define also R1

  • It has been corrected and R1 have been added.

page 22: 3-mercaptopropinoic, correct

  • It has been corrected.

page 23, rows 11-12: add the verb

  • It has been added.

page 25, 76-91%

  • It has been corrected.

Scheme 50: add nitro group in formula 177

  • It has been added.

Scheme 51: hexcyl?

  • It has been corrected to "hexyl"

Scheme 53: flourobenzyl?

  • It has been corrected to "fluorobenzyl"

Scheme 65: therephtaldehyde

  • I has been corrected to " terephthalaldehyde"

Page 35: quinoline derivative 222

  • It has been corrected as " Quinoline derivative 222"

Page 38, row 2: is not an hydrolysis!

  • It has been changed to "dehydration"

References: publication year and journal name abbreviations must be checked carefully

  • The references have been checked

Moreover, along the text, many errors on verbs and plural/singular words must be corrected.

  • It has been corrected.

Reviewer 2 Report

This is a good review and I can see that the authors have made some some edits. However, there is still more work to do and the authors while saying they have addressed previous comments unfortunately they have not. The edits in the text are also not fully highlighted (not an issue if all were done). There are quite a few small errors in this paper, it needs to be properly proofread.

There are several examples of not just bad English but poor sentence structure.

eg Line 21/22 '...itself which was encouraged us to shed new light...'

There are also alot of small errors that detract from an otherwise very good paper -

Line 23/24 the keyword template is not removed.

Line 166, 203, 211 - font error 

This is not exhaustive, please re-read the work.

- The abstract needs to be re-written to be coherent.

Line 29/30 'These types of heterocycles constructed a large number of drugs...' these drugs were not constructed. It should read 'These heterocycles form integral scaffolds for a large number of drugs...'

line 45 what evidence do the authors have that this compound is 'currently undergoing' trails etc? The scaffold has been shown to have potent anti-HIV/AIDS activity, it needs to be written like this. In the response to this reviewer, it says this has been corrected - it hasn't. This is actually an interesting observation and doesn't need to be over-sold. 

The conclusion sentences don’t really add much in the current format. It would be better if the authors provided their insights into this scaffold rather than just list off the schemes. This is a very interesting scaffold and the authors should highlight this.

The errors are just examples, the authors must proofread this work.

Author Response

Response to reviewer 2

This is a comprehensive review of the field of thiazinane and does bring together a lot of information in one place. It is definitely valuable to the field and a good read for anyone interested in this scaffold. However, before it can be published quite a lot of mainly small things need to be addressed. These come together to detract from what is overall a nice summary of the current state of the field.

- In this sense it would be nice to see that authors provide atleast a full paragraph (or more) to the conclusion section to highlight some of the key features summarised in this review. If this was done, the review would have much more impact.

  • A paragraph was added to the conclusion.

- The whole paper needs to be thoroughly proof read and corrected including for English and grammar.

  • The paper has been checked for grammar. 

- The abstract while mostly technically correct suffers from bad English and needs to be proof read eg. 'which used for AIDS treatment' should be 'which has been used for AIDS treatment' etc

  • The abstract has been changed..

Specifically on the subject of ref 5 (10.1073/pnas.0402357101), this compound has been demonstrated to an ability to be used as an AIDS treatment. This is not proven so far as I see no human patients were involved in this reference, so it has been shown to have anti-HIV activity by a mechanism that should work as an anti-AIDS. This needs to be changed in the abstract and text otherwise the reader could be mis-led. This also needs to be addressed in the main text as there is no evidence that there is any ‘current’ work on this series. That fact that that functionality was used in a medchem program is interesting anyway and does not need to be over sold.

  • It has been corrected.

Why is this the structure of 5-(1,1-Dioxido-1,2-thiazinan-2-yl)-N-(4-fluorobenzyl)-8-hydroxy-1,6-naphthyridine-7-carboxamide and 5-(1,1-Dioxido-1,2-thiazinan-2-yl)-N-(4-fluorobenzyl)-8-hydroxy-1,6-naphthyridine-7-carboxamide not present in this introduction despite been mentioned in the abstract and main text.

  • The structure has been inserted in figure 3 with its name.

More details should be provided in the introduction to provide more context to the use of the thiazinane

  • Some details have been inserted and highlighted with yellow.

- Consistency and Errors

There are many small errors contained in this review eg figure 2

eg bretschneiderazines A not fully highlighting the 3-thiazine functionality as blue. This is a consistent error where in some cases not enough is highlighted an later on eg Scheme 87 the whole oxindole is highlighted aswell (stick to just the functionality)

  • Corrections have been made, and only thiazinane ring highlighted blue along the paper.

There is a lack of consistency of functional group representations nitro and aldehyde eg figure 4 vs Scheme 74 (aldehyde) and figure 4 vs Scheme 37 (nitro)

  • The nitro as well as the aldehydic groups have been changed .

This happens later in the text with SO2 vs O=S=O, there needs to be consistency. I personally prefer the correct angled O=S=O 

  • It has been changed.

- Examples of smaller errors - there are more not highlighted here

Figure 4 should have the names with a capital letter or not.

  • It has been corrected.

Scheme 5 - remove 'stirring' and throughout text unless really required eg Scheme 6, 61, 67 etc

  • It has been removed.

end of page 6 'OTF' should be 'OTf'

  • It has been corrected.

check scheme 12 for the correct representation

  • It has been checked and corrected.

Scheme 25 needs to show all substrates on scheme

  • The substrates have been inserted on the scheme.

Scheme 29 needs cleaning up, move CO2H to the lower position and clean up sizes

  • The structure has been cleaned up.

Scheme 35 n-butyl - 'n' should be in italics

  • It has been corrected

132 and 133 should be cleaned up and have an accurate representation

  • They have been cleaned.

Mumm rearrangement needs to cite - Mumm, O. Ber. Dtsch. Chem. Ges. 1910, 43, 886-893

  • It has been inserted as reference [77]

Scheme 53. Issue with capital consistency

  • It has been checked and corrected.

4-[[40-(bis(2-hydroxyethyl)amino)phenyl-10-]azo]-3-nitrobenzaldehyde 'bis' should be italics

  • It has been corrected.

Scheme 75 and Scheme 97 need to be cleaned up, errors and alignment

  • They have been cleaned up and readjusted .

Scheme 98 is not aligned correctly and properly spaced out.

  • It has been readjusted.
